# Study on Flow Characteristics of Working Medium in Microchannel Simulated by Porous Media Model

**DOI:** 10.3390/mi12010018

**Published:** 2020-12-26

**Authors:** Yufan Xue, Chunsheng Guo, Xiaoxiao Gu, Yanfeng Xu, Lihong Xue, Han Lin

**Affiliations:** 1School of Mechanical, Electrical & Information Engineering, Shandong University, Weihai 264209, Shandong, China; xueyufan1129@163.com (Y.X.); ocjb6234@163.com (X.G.); 201916564@mail.sdu.edu.cn (Y.X.); 202017437@mail.sdu.edu.cn (L.X.); 2Centre for Translational Atomaterials, Faculty of Science, Engineering and Technology, Swinburne University of Technology, P.O. Box 218, Hawthorn, VIC 3122, Australia; hanlin@swin.edu.au

**Keywords:** microchannel array, porous media, numerical simulation, theoretical research, experimental validation

## Abstract

As a phase change evaporator, a microchannel array heat exchanger is of great significance in the field of microscale heat dissipation. The performance of which strongly depends on the flow resistance, capillary force, and other factors. In order to improve the heat dissipation efficiency, it is necessary to perform an in-depth study of the characteristics of microchannel flow using numerical simulation. However, the current simulation model requires high computational cost and long simulation time. To solve this problem, this paper simplifies the numerical simulation of the rectangular parallel array microchannels by building the basic flow model based on the concept of porous media. In addition, we explore the effect of aspect-ratio (AR), hydraulic diameter, inlet velocity, and other parameters of fluid flow behavior inside the microchannels. Meanwhile, a user-defined function (UDF) is formulated to add the capillary force into the model to introduce capillary force into the porous media model. Through the above research, the paper establishes the porous media model for single-phase and gas-liquid two-phase flow, which acts as a simplification of microchannel array simulation without grossly affecting the results obtained. In addition, we designed and manufactured experiments using silicon-based microchannel heat exchangers with different-ratios, and combined with the visualization method to measure the performance of the device and compared them with simulation results. The theoretical model is verified through the suction experiment of array microchannel evaporator capillary core. The simplified model of microchannel array significantly saves the computational cost and time, and provides guidance for the related experimental researches.

## 1. Introduction

With the development of information technology and the advent of the 5G era, electronic components are evolving in terms of high-frequency, high-speed, and integrated circuits becoming miniaturized. The heat flux density of electronic components also rises dramatically, and the development of mobile terminals also puts forward higher requirements on the size of electronic cooling components [1]. The research shows that a flat heat pipe can reduce the temperature of the chip to the maximum extent, increase its overall temperature uniformity, and take very small heat dissipation space, which has significant advantages in high intensity heat dissipation and temperature control for micro-scale components [2]. At present, extensive research has focused on flat heat pipes (FHP) and vapor chambers (VC) [3]. The former is made by directly flattening the traditional cylindrical heat pipe, which is composed of evaporator, condenser, and adiabatic part [4]. The latter is an airtight container with a capillary-core structure covered on the inner wall and a condenser and evaporator on both sides [5]. However, in recent years, microchannel array heat exchangers have attracted more attention [6]. The microchannel array heat exchanger has a small overall size, and the key component of its heat dissipation is the array microchannel, featuring a high surface area to volume ratio and a small overall volume. In addition, the microchannel has a small groove width and a high aspect ratio for providing self-driven capillary force. Accordingly, it has advantages over commonly used heat exchangers and becomes a promising candidate for a compact heat exchanger. There are single-phase and two-phase liquid flows in the evaporative heat dissipation of the microchannel heat exchanger. The flow resistance and capillary force determine the performance of the fluid flows in the microchannel, which affects the supply of working medium and evaporation rate in the heat exchanger. Those parameters further determine the heat dissipation efficiency of the heat exchanger. Therefore, this paper aims to simplify the previous model. The calculation time is shortened, and the accuracy is verified by experiments. It is of great significance to explore the research method of flow characteristics of microchannel array for further exploiting the technological advantages and meeting the requirements of microelectronic heat dissipation [7].

The microchannel heat exchanger, which combines electronic components to realize two-phase dissipated heat in limited space, has been widely concerned. The microchannel heat exchanger requires both the small scale and heat dissipation effect. Many scholars have carried out detailed studies on the key factors of microchannel heat exchanger in affecting the performance. Erp [8] designed a high-performance silicon microchannel heat exchanger, which realized high heat dissipation efficiency and high heat flux, and achieved energy saving. They confirmed the feasibility and value of silicon microchannel heat exchanger. For the numerical simulation of microchannel array, David O [9] numerically studied the effect of structural parameters on the performance and the advantages of the rectangular parallel array microchannel. Sahar A M [10] used FLUENT to simulate single-phase heat transfer in a copper rectangular microchannel and verified that the uniform flow distribution can be realized with the decreasing inlet pipe diameter. Jiang P X [11] compared the measured performance of microchannels and porous media micro heat exchangers using numerical simulation. However, in previous studies, only a transverse comparison has been made between the microchannel and porous media heat exchangers. The above-mentioned numerical studies without any simplifications result in complex processes and large computational cost, making it unsuitable for simulating a heater exchanger with hundreds of microchannels. Wang C Y et al. [12] used a single capillary model to estimate the flow behavior in porous media and modified the capillary pressure formula. In addition, they demonstrated the important role of capillary forces in fluid flow. In this way, they formulated the theoretical model of fluid flow and heat transfer in micro heat pipe with diameter of 0.1–0.3 mm by using the method for porous media. However, they studied a single capillary tube at millimeter scale, which cannot be directly applied to the array microchannels. This method used porous media model to capillary simulation, which greatly simplified the numerical modeling of array microchannels. It should be noted that a novel idea was provided based on this study that by using a method coupled porous media model to capillary simulation the numerical modeling of array microchannels can be greatly simplified.

The microchannel heat exchanger has the advantages of small size and large capillary suction [13,14,15]. The gas-liquid two-phase microchannel has been regarded as a promising pathway for heat dissipation. Therefore, it is necessary to perform numerical and experimental investigations on its flow performance. One bottleneck issue in the numerical study is the high computational cost and long computing time for simulating the models with the large number of channels. To solve this issue, the porous media method is applied to significantly simplify the model and save computational cost and time, without compromising the accuracy. In this paper, we study the influences of parameters such as permeability, pressure, suction distance, and other parameters on the flow in microchannels. The maximum heat dissipation efficiency of microchannel heat exchangers with different parameters are different because the aspect ratio affects the capillary force. These factors directly affect the flow, and flow affects liquid replenishment, reflux, etc., thus affecting heat dissipation.

This paper studies a high-aspect-ratio silicon based rectangular microchannel array heat exchanger with the number of microchannels between 67 and 167. In the numerical simulation, the mesh size is set to less than one micrometer for high accuracy. In order to simplify the simulation process and save computational cost, this study proposes an innovative concept of “substitute simulation” to establish a porous media model. The proposed model is simple and accurate, and useful to numerically simulate microchannel array, which can be used to design and understand experiments.

## 2. Establishment of Theoretical Model of Single-Phase Flow Porous Media

This paper simulates a silicon-based rectangular microchannel array with different aspect-ratios, equivalent diameters, and inlet flow velocities using FLUENT. The aspect-ratio is the ratio of the cross-sectional depth to the width of the channel. Then we modify the porous media momentum equation to simulate the internal working medium flow pressure drop loss. As a result, the porous media model can replace the conventional microchannel array model to simulate the internal working medium flow.

### 2.1. Physical Model

The main structure of the microchannel heat exchanger is shown in Figure 1, which is composed of five parallel microchannels, with each single microchannel of 4 mm in length, 180 μm in depth and 30, 20, 16, 12 μm in width. Channel spacing is the same as channel width. Mesh software is used to generate uniform hexahedral mesh. The porous media part is a cuboid with the same size as the five microchannel array.

In this paper, the Volume of Fluid (VOF) model was selected to simulate the flow inside the microchannels and SIMPLE algorithm was used to solve the pressure-velocity coupling equation. As the channel size was small and the flow rate was slow, the working medium inside the channel was a laminar flow, which was activated for calculation. The porous media model was simulated. Porosity was a key parameter, which is the percentage of the pore volume in a porous medium and the total volume of the material. The porosity was set to 0.5 because the width of the microchannel is equal to that of the interval part.

The following assumptions were made in the numerical simulation:Stable fluid flow;Incompressible fluid;Laminar flow;The solid and fluid properties are constant except the viscosity of water;

### 2.2. Mathematical Model

The governing equations of incompressible fluid can be derived from the conservation of mass and momentum. In the case of steady flow, the following equation can be obtained:

Continuity equation:(1)$$\frac{\partial \rho }{\partial t}+\frac{\partial }{\partial x}\left(\rho \upsilon \right)=0$$

Momentum conservation equation:(2)$$\rho \frac{\partial \upsilon }{\partial t}=-\nabla p+\rho F+\mu \Delta \upsilon$$
where *ρ* is the density of the fluid, ***υ*** is the velocity vector of the fluid, ***P*** is the isotropic pressure of the fluid, ***F*** is the volume force, and *μ* is the dynamic viscosity.

### 2.3. Boundary Conditions

Inside the silicon-based rectangular microchannels, the fluid working medium is water. Here we use the velocity inlet, the pressure outlet and no slip boundary conditions. The inlet boundary conditions are from 0.01 m/s to 0.03 m/s, and the outlet pressure is one atmosphere (1.01 × 10^5^ Pa). The parameters of fluid and channel material are shown in the Table 1 below.

In order to establish the geometric model of porous media, FLUENT simulating a specific fluid domain is used, instead of a real solid area, and we added the consumption of the momentum source term to consider the impact of porous media. Given the pressure loss, it could be calculated using the nominal velocity. In experiments, parameters of porous media can be calculated using experimentally measured pressure and velocity.

The empirical formula of the resistance of the porous media model was combined with the model region. Porous media were modelled by adding a momentum source term to the standard fluid flow equations as [16]:(3)$$S=-\left(\frac{\mu }{K}\upsilon +C_{2}\frac{1}{2}\rho \left|\upsilon \right|\upsilon \right)$$
where, the first term is the viscous loss term, the second term is the internal loss term, K is the permeability, and *C*_2_ is the inertial resistance factor. The source term in the porous media theory was modified to simplify the numerical simulation of microchannel arrays, which can demonstrate results that agree well with the actual microchannel simulation results.

Darcy’s law is generally used to describe the flow characteristics of liquid with a Reynold number less than 10 within porous media [17]. Here, the flow obeys Darcy’s Law, which is significantly affected by the viscous force. Qu [18] also confirmed that viscous force is the key factor of flow at micro scale. Therefore, it can be seen from theoretical Equation (3) that the permeability in the viscous loss term inside the microchannel is modified to derive the empirical formula of the viscous loss coefficient. Then, the viscous resistance coefficient can be set in FLUENT to simulate the effect on microchannel array.

The viscous loss of fluid was derived from the permeability of fluid flow in a small space, which affects the pressure loss of fluid flow. The permeability in the fluid flow was calculated as the resistance coefficient of the viscous loss term in the porous media model [19]. Since Darcy’s law is derived from Hagen-Poisseuille’s theorem, the paper makes a comparative study of fluid flow in porous media and microtube bundles. The microbundle model with equal diameter is used to describe porous media, and Darcy’s Law is verified by the Hagen-Poisseuille theorem [20], Hence, the permeability equation is expressed as follows:(4)$$K=\frac{\varphi }{8}r^{2}$$
where φ for porosity and r for pipeline radius.

In the transport model of porous media, the flow law of working medium can be expressed using Darcy’s Law as:(5)$$Q=\frac{AK\Delta P}{\mu H}$$
where, *Q* is the volume flow rate, *K* is the permeability, *A* and *H* respectively refers to the sample’s cross-sectional area and characteristic length, Δ*P* refers to the pressure difference at both ends of the fluid flow path, μ is the fluid viscosity.

Pressure drop was obtained through numerical simulation at different inlet velocities to obtain pressure drop at inlet and outlet to study flow resistance [21,22] and the corresponding permeability was calculated through Equation (5). Geometric models with different aspect-ratios were simulated respectively. The porosity of microchannel was 0.5, and the water intake force radius was substituted into Equation (4) to calculate the permeability of microchannels with different aspect-ratios. The results are shown in the Figure 2.

However, there were some errors due to the difference in channel arrangement and cross section. The aspect-ration is introduced as a parameter to correct the errors through regression analysis. The capillary permeability formula for calculating rectangular cross section microchannel beam permeability is:(6)$$K=\varphi {\left(\frac{A}{P^{\prime }}\right)}^{2}\cdot AR^{-0.03}$$

The simulation of cases with different ARs were performed to verify Equation (6), as shown in Figure 3. It showed a good agreement between the simulated results and that calculated using Equation (6) with an error below than 4%. Therefore, the model could be used to simulate the flow resistance inside the microchannel.

By modifying the important parameters of the resistance term in the momentum equation of the porous media model, the permeability calculation formula in the single-phase flow model of the porous media was obtained. Thus, it is possible to simplify the simulation of the microchannel with the same external contour size and boundary conditions. The pressure drop of the inlet and outlet cross-section from different Ars from the two models are compared and shown in Figure 4.

It can be seen that difference of the pressure drops from the porous media and the microchannel models was with 15%, which confirmed the accuracy of the porous media model.

## 3. Establishment of Two-Phase Flow Porous Media Theoretical Model

The flow characteristics and heat transfer performance of two-phase flow in microchannels are quite different from channels with large size. The two-phase fluid flow in the microchannel was not only influenced by the viscous stress, but also strongly depended on the surface tension [23]. The surface tension of the working medium drove the liquid in the microchannel, which can improve the efficiency of microchannel heat exchanger [24]. Thus, research on the capillary force is of great importance [25].

### 3.1. Physical Model

In this section, the gas-liquid two-phase flow of rectangular parallel microchannel is numerically simulated. The geometric models of closed three-dimensional (3D) parallel array microchannels with different ARs and porous media with porosity of 0.5 were established. The capillary force was calculated in a 4-mm long rectangular area, which in different models was respectively micro channel and porous medium. The microchannel had a depth of 180 μm and a width of between 20 and 300 μm, which resulted in different aspect-ratios. The geometric model was a closed loop. The initial state setting of gas-liquid two-phase distribution is shown in the Figure 5.

### 3.2. Mathematical Model

The working medium of microchannel two-phase flow was deionized water and air, and gas-liquid two-phase was incompressible. The VOF multiphase flow model was used with the continuum surface force model (CSF) introduced to consider the effect of surface tension.

Volume fraction equation of VOF model:

If the volume fraction of the *k* phase fluid is set to $\alpha_{k}$, then the relationship between the VOF equation of the *k* phase and each volume fraction *α_k_* is:(7)$$\frac{\partial \alpha_{k}}{\partial t}+\alpha_{k}\nabla \cdot V=S_{\alpha_{k}}$$
(8)$$\overset{n}{\underset{k=1}{\sum }}\alpha_{k}=1$$
where $S_{\alpha_{k}}$ denotes the mass exchange between phases. In this paper, the mass exchange between phases is not considered, so the source term at the right end of the equation $\;S_{\alpha_{k\;}}$ is zero.

Continuity equation of VOF model:(9)$$\frac{\partial \rho }{\partial t}+\rho \nabla \cdot V=0$$
where ρ is the density.

Momentum equation of VOF model:

The momentum equation in the VOF model was used in the entire flow field region, and the velocity field obtained by solving was used by all phases, in the following form:(10)$$\frac{\partial }{\partial t}\rho V+\nabla \cdot \rho VV=-\nabla p+\nabla \cdot \left[\mu \left(\nabla V+\nabla V^{T}\right)\right]+F+\rho g+S$$

Density ρ and dynamic viscosity  μ are the basic properties of the fluid, *P* is the stress tensor acting on the surface per unit volume, ***F*** is the mass force per unit volume, and S is the source term, including the seepage resistance and capillary suction.

Continuum surface force model (CSF):

In FLUENT, the CSF model was proposed by Brackbill [26], in which the governing equation added a source term of the surface tension. The pressure drop across the two-phase interface depended on the surface tension coefficient $\sigma_{c}$ and the surface curvature *R*_1_ and *R*_2_ measured by the orthogonal directions of the two radii:(11)$$p_{2}-p_{1}=\sigma_{c}\left(\frac{1}{R_{1}}+\frac{1}{R_{2}}\right)$$
where, $p_{1}$ and $\;p_{1}\;$ represent the pressure on both sides of the interface.

### 3.3. Theoretical Derivation

Firstly, the capillary force of two-phase flow was simulated to study the relationship between capillary force and microchannel size.

The suction distance of liquid in the microchannel was used to characterize the capillary force, which was difference of the ordinate of the monitoring point at the center of the liquid level in the microchannel and porous medium. Meanwhile, the microchannels with different ARs were simulated for comparison. The corresponding relationship between the maximum suction distance and aspect-ratio (AR) of each microchannel is shown in Figure 6, in which one can see that the capillary suction force of microchannels with different ARs are different.

The relationship between the suction distance and the ARs of the microchannel was obtained through data fitting, and a deviation of less than 15% was observed compared with the numerical simulation. Therefore, Equation (12) could be used to calculate the ARs of the microchannel and the suction distance.
(12)$$h=\left(-9.58e^{-\frac{AR}{1.05}}+5.63\right)\times 10^{-4}$$

In the numerical simulation of two-phase flow with porous media instead of microchannels, the effect of capillary force was considered as the focus of the research, which has also been highlighted in the simulative experiment of capillary core. Capillary force was produced by the attraction between liquid molecules and the difference between external forces on liquid molecules. The force that causes the surface of a liquid to contract was surface tension. The CSF model was used in FLUENT to add the surface tension source term for the fluid in multiphase flow. The contact angle between the liquid and the wall was used to simulate the capillary effect of the fluid. However, the CSF model was actually unable to represent the effect of surface tension within the porous media, because the porous media is modelled as a pseudo-fluid area rather than solid in the simulation. The fluid flow in porous media was simulated by including the resistance source term and related parameters. Therefore, in this case the capillary suction fluid in porous media cannot be modeled by a CSF model. Another model to simulate capillary suction is needed.

Capillary force takes place at the gas-liquid interface. In FLUENT, the gas-liquid interface was allocated on a small computing cell. A user-defined function (UDF) was written by deriving the representation of capillary force in the cell to add the capillary force. The method of FLUENT was the finite volume method. In order to meet the calculation requirements, it was necessary to transform all non-volume integrals into volume integrals, then carry out differential calculation, and finally discretize the entire definition domain. Taking the general momentum equation for example, by assuming a volume of fluid inside the cell is Ω, the cell boundary as S, the rate of change of momentum is equal to the quality of the force and the sum of surface force acting on the unit. The quality of the force is expressed as:(13)$$\overset{}{\underset{\Omega }{∭}}\rho Fd\tau$$
where *F* is the quality of power, *ρ* is fluid density, and surface force is:(14)$$\overset{}{\underset{\Omega }{\iint }}p_{n}dS=\overset{}{\underset{\Omega }{\iint }}n\cdot PdS$$

On unit area of surface force $p_{n}=n\cdot P$, ***n*** as a unit vector along the normal direction, *P* is second-order symmetric stress tensor, which needs to transform the area of the surface of the integral force into the volume integral force. Here we use the Gauss divergence theorem,
(15)$$\overset{}{\underset{\Omega }{\iint }}n\cdot PdS\overset{}{\Leftrightarrow }\overset{}{\underset{\Omega }{∭}}divPd\tau$$

*div**P* is the divergence operation of *P*, so that the rate of change of momentum can be written as:(16)$$\frac{D}{Dt}\overset{}{\underset{\Omega }{∭}}\rho Fd\tau =\overset{}{\underset{\Omega }{∭}}\rho Fd\tau +\overset{}{\underset{\Omega }{∭}}divPd\tau$$

All of these terms are converted into integrals with respect to volume, after modifying the form, the differential form of the momentum equation can be obtained.

Capillary force suction can also be regarded as a surface force acting on a unit near the interface between liquid and gas, which therefore is also applicable to the Gauss divergence theorem. We can convert the capillary force as follows:(17)$$2\overset{}{\underset{\Omega^{\prime }}{\iint }}n\cdot P_{c}dS\overset{}{\Leftrightarrow }2\overset{}{\underset{\Omega^{\prime }}{∭}}divP_{c}d\tau$$
where $P_{c}$ is the capillary pressure at the two-phase interface, which is integrated twice, because the surface tension is acting on both the inner and outer surfaces of the liquid cell. $\Omega^{\prime }$ is for the whole domain of the unit cell. Since surface tension only exists at the gas-liquid interface, the equation should be further transformed to conform to this characteristic. On this basis, the above capillary force equation is adapted.

We add the liquid volume fraction $T_{l}$ in the Equation (17) of integral item. In order to simplify the calculation, capillary pressure $P_{c}$ for constant force is assumed to be along the axial direction of porous media, and $\;T_{l}$ is related to space coordinates with a scalar value between 0 and 1. The equation can be reduced to the following form after the divergence of the terms on the right of the equals sign:(18)$$2\overset{}{\underset{\Omega }{\iint }}n\cdot P_{c}T_{l}dS\overset{}{\Leftrightarrow }2\overset{}{\underset{\Omega }{∭}}P_{c}T_{l}\nabla T_{l}d\tau$$

It is important to note that the formula ∇$T_{l}$ is a gradient of the liquid phase volume fraction in the cell, rather than divergence, because the liquid phase volume fraction $T_{l}\;$ is a scalar function instead of a vector function.

The volume fraction gradient ∇$T_{l}\;$ can show the gas-liquid interface changes, so, this formula can precisely represent the property that surface tension exists only at the gas-liquid interface, it also confirms the correctness of this new model, which characterizes capillary suction force of porous media, whose capillary suction final form of the source term is the following [27]:(19)$$F_{y}=2P_{c}T_{l}\nabla T_{l}$$

According to Equation (19), there are two terms in the UDF of capillary suction, namely, phase volume fraction and phase volume fraction gradient. In the VOF model, there may be multiphase in a grid, so when extracting parameters such as volume fraction in the UDF, which can have different values in the same cell, the ordinary pointer cannot be used directly, but the super domain pointing to its sub-structure can be used instead.

Additional UDF need to be written to extract the volume fraction gradient indirectly. The basic idea of the UDF program is to set up user-defined scalar UDS and memory UDM. Before the beginning of each iteration step, the UDF extracts the volume fraction of the whole domain T_l_ and saves in the UDS. Then the gradient of UDS is calculated to get the volume fraction gradient ∇T_l_, which is saved in the UDM. Finally, the capillary force source term is calculated according to the UDS and UDM. Through calculating the value of $P_{c}$, the UDF gets the relation between the capillary suction force and the liquid volume fraction. This relation is written as UDF and added to the momentum source term of porous media. In this way, the capillary force can be introduced into the porous media model.

However, UDF needs to represent the capillary forces of microchannels with different ARs, so the capillary pressure $P_{c\;}$ is modified according to the microchannels. This paper explores the relationship between $P_{c}$ value and microchannel related parameters. In addition, it introduces different size parameters into the UDF to represent the capillary force, to achieve the effect of the capillary force of microchannels with different ARs. Therefore, the capillary suction process of porous media was numerically simulated, the capillary force UDF was added, and the value of $P_{c}$ was properly modified. The corresponding suction distance was recorded at the same time for comparison.

From the simulation results, it can be seen in Figure 7 that the *P_c_* value changes with the suction height in a linear relationship as,
(20)$$P_{c}=596500\cdot h-256$$

However, the current UDF does not show the difference in capillary force of microchannels with different ARs. Therefore, by introducing the AR to modify the capillary force formula and combining it with the microchannel suction distance, the relationship between the microchannel AR and *P_c_* value is formulated as follows:(21)$$P_{c}=-571.4e^{-\frac{AR}{1.05}}+79.8$$

Formula (21) is used to calculate the *P_c_* values corresponding to different ARs. The obtained suction distance was compared with the results calculated by the microchannel geometric model. The results are shown in the Figure 8 below.

The results showed an error of less than 10%. Therefore, when the microchannel is gas-liquid two-phase flow, the UDF representing capillary force can be added, and the corresponding parameters can be changed according to the AR of microchannel.

Therefore, the flow theoretical model of porous media instead of microchannel simulation was successfully established.

Permeability correction formula:(22)$$K=\varphi {\left(\frac{A}{P}\right)}^{2}\cdot AR^{-0.03}$$

$P_{c}\;$ value correction formula in the capillary force UDF:(23)$$P_{c}=-571.4e^{-\frac{AR}{1.05}}+79.8$$

This theoretical model was based on changing the values of *K* and *P_c_* to change the flow resistance in the porous media model and add capillarity, which makes the fluid flow force consistent with the microchannel model. Thus, the porous media model can be used to simulate the rectangular parallel array microchannels with high aspect-ratios replacing the microchannel model. In order to further verify the correctness of the numerical simulation model, the experiment was carried out.

## 4. Capillary Suction Experiment of Microchannel Heat Exchanger

The capillary force suction process of liquid in microchannels was studied experimentally by using a home-made silicon-based heat exchanger with different ARs. The experimental results were recorded to explore the influence of AR on capillary force, and the theoretical model was validated by comparing the simulation results with the experimental results.

Considering the Deep Reactive Ion Etching (DRIE) processing technology and structural stability, the upper limit of the aspect ratio is 15:1, and the minimum width is 12 μm. Four types of silicon microchannel evaporator were fabricated, with a depth of 180 μm, and with widths of 12, 16, 20, 30 μm, respectively. In order to visualize the experiments at operating temperatures, we used silicon-based motherboard with Pyrex 7740 glass cover bonding [28]. The silicon-based motherboard comprised of a microchannel array and a working medium inlet and outlet channel, and an inlet and an outlet were arranged on both sides for charging working medium. The microchannel array evaporator had an area of 4 × 4 mm in the middle. The STS-DRIE machine was used to etch the silicon substrate to a depth of 180 μm. Then, the etched silicon wafer was thinned and polished by AM HRG-150 thinning machine and AMASP400 grinding and polishing machine. The DRIE can perform anisotropic processing on silicon wafers by process adjustment to achieve high-aspect-ratio. Finally, the KARLSUSS SB6 bonding machine was used to package the silicon substrate and the Pyrex upper cover through anodic bonding technology. The experimental schematic diagram is shown in the Figure 9 and Figure 10 below.

The visual suction experimental system is shown in Figure 10. The experimental system consisted of the holding module, visual module, and data acquisition module. The deionized water was selected as the working medium. In the initial state, the water inside the microchannel was drained and pushed into the space in front of the evaporator through a micro-fluidic syringe. When the working medium contacted the microchannel entrance, the capillary force produced suction phenomenon, as shown in Figure 11. The Photron Fastcam NOVA S12 high-speed camera was used to capture the liquid flow in the microchannel and measure velocity. The average suction distance in the corresponding time could be calculated.

As can be seen in Figure 11, the pumping distances of the liquid were not uniformly distributed. This phenomenon can be explained as follow: the inlet pipe diameter and the overall size of the heat exchanger were small, making liquid not enter the cavity in front of the array microchannel uniformly, thus resulting in a time difference in suction phenomenon. Furthermore, due to the uncertainty of the thrust force of the syringe, as well as the influence of measurement error, a single experiment had certain fluctuations. Therefore, the experiments on microchannel of every AR were repeated five times. By filming the full process using the high-speed camera, the suction speed of each microchannel with effective flow was measured, and the average value was calculated.

The average suction speed of deionized-water in microchannels with four different ARs were 0.033, 0.032, 0.03, 0.025 m/s, respectively, which were compared to simulation results.

It can be seen in Figure 12 that the experimental values were slightly higher than the numerical simulation. This was due to the fact that the thrust of the manual operation is difficult to eliminate during the experiment, as a result, the final capillary suction phenomenon may be mixed with a certain thrust. Nevertheless, the difference between the experimental and simulation results remained within three percent and the overall trend was consistent. Therefore, it is considered that this porous media theoretical model can replace the microchannel model within the range of 5 to 15 of AR as shown in this study.

## 5. Conclusions

In this study, a new theoretical model of porous media is established to simplify the directly simulation of single-phase flow and two-phase flow in array microchannel. The main conclusions are as follows:Parameters such as porosity, AR and hydraulic diameter are introduced to correct the viscous loss term in the momentum equation of the theoretical model of porous media. A single-phase flow theoretical model of porous media was used to simplify the microchannel simulation using this modified viscous loss approach.Through the theoretical study of capillary forces in porous media, a UDF representing capillary forces is written, and parameters associated with the AR of the microchannel are introduced to relate the capillary forces and AR related. Accordingly, another porous media model used for simulating the two-phase flow process in microchannels was setup.We designed the overall size of the three different ARs for a 4 × 4-mm microchannel array evaporator. The microchannel widths were 12, 20, 30 μm, and the depth was 180 μm. The microchannel suction experiment was visually recorded to verify the theoretical model.

The results of this study are applicable to the numerical simulation of microchannel in a certain AR range of 5 to 15, which provides a basis for optimizing the numerical simulation of microchannel-based structures. Meanwhile, it can be further extended by introducing other more complex factors. It is useful for designing and predicting experiments of array microchannel. The porous media model greatly simplifies the numerical simulation process of rectangular array microchannel.

## Figures and Tables

**Figure 1 micromachines-12-00018-f001:**
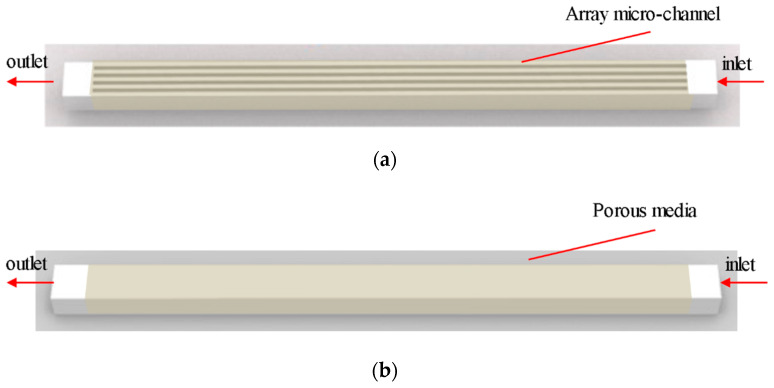
Schematic of the single-phase flow model: (**a**) The brown part of the figure is a microchannel; (**b**) The brown part of the figure is porous media. The inlet and outlet are indicated by the arrows.

**Figure 2 micromachines-12-00018-f002:**
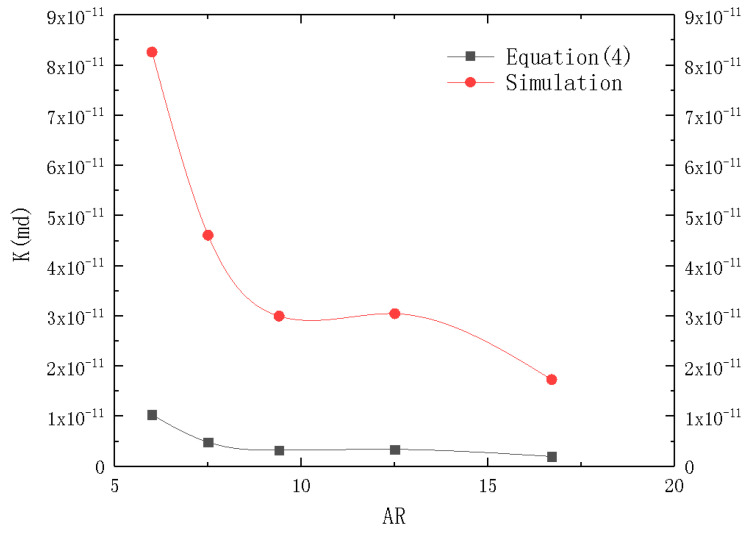
Plot of permeability (K) depending on aspect-ratio (AR).

**Figure 3 micromachines-12-00018-f003:**
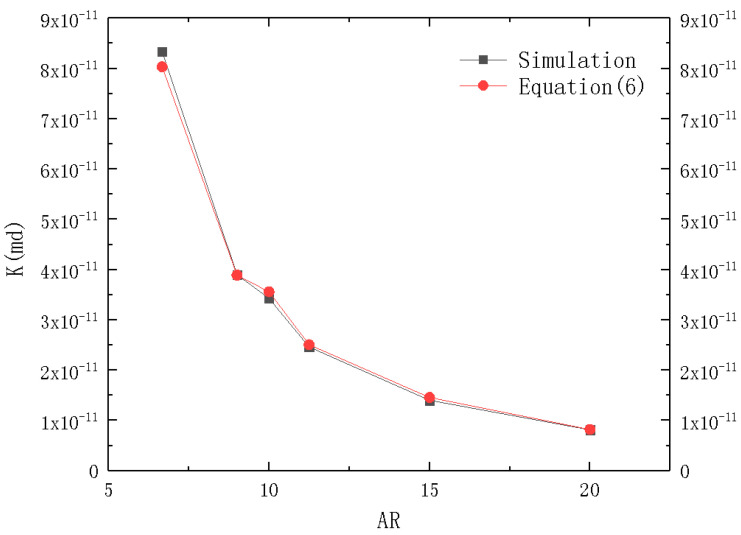
Comparison of permeability Equation (6) (black squares) and simulation results (red circles).

**Figure 4 micromachines-12-00018-f004:**
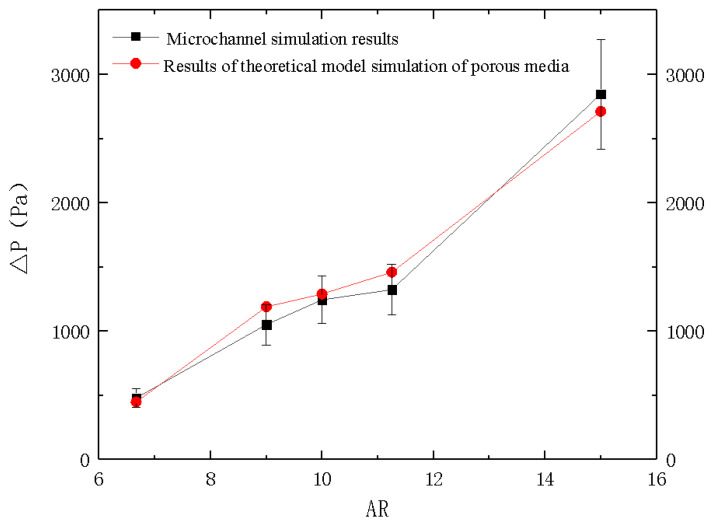
Curves of pressure drop on different ARs from microchannel and porous media simulations.

**Figure 5 micromachines-12-00018-f005:**
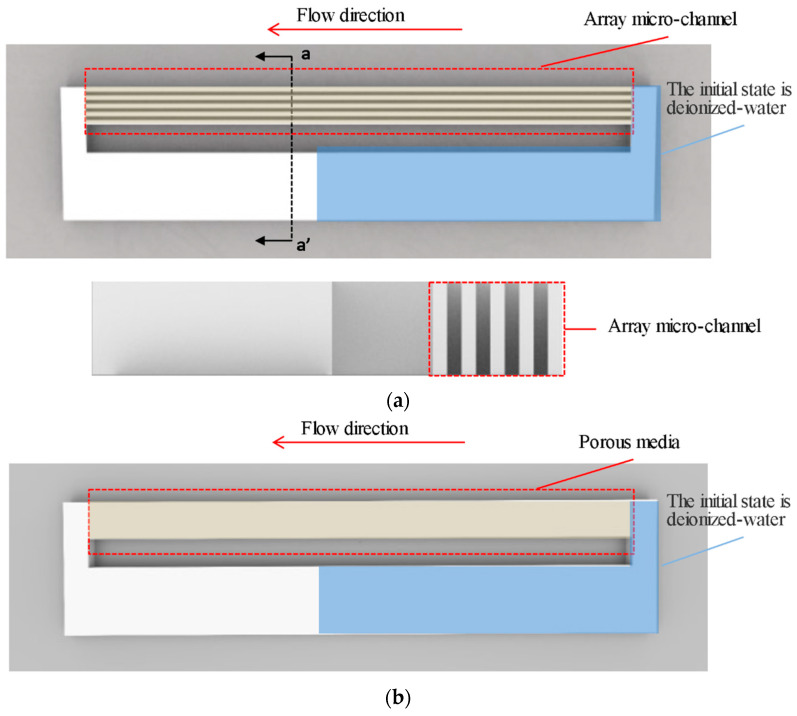
Physical models of two-phase flow: (**a**) The physical model of two-phase flow includes a cavity part and a microchannel part. The blue part is deionized-water and the rest is air at the initial state. The direction of liquid suction is indicated by the arrow; (**b**) The physical model of two-phase flow of porous media is basically the same as that of microchannels with the microchannels partially replaced by porous media.

**Figure 6 micromachines-12-00018-f006:**
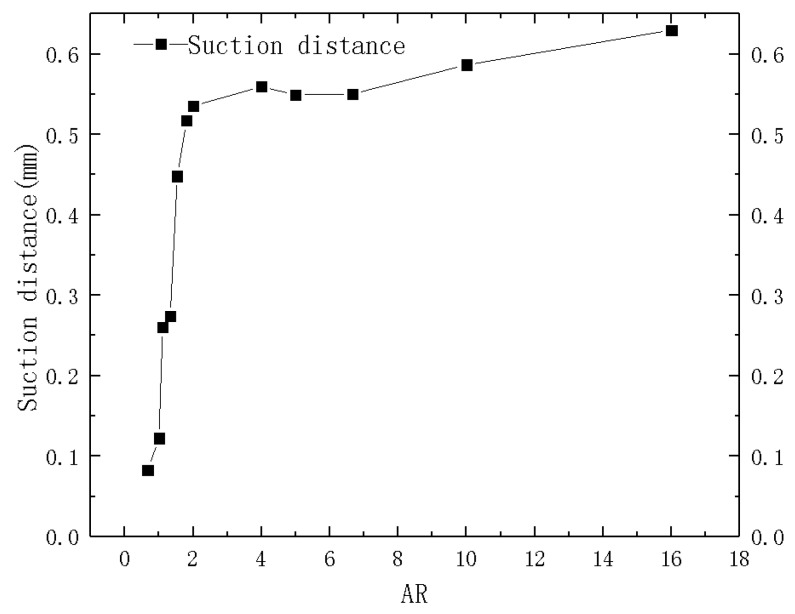
Microchannel suction distance changing with ARs.

**Figure 7 micromachines-12-00018-f007:**
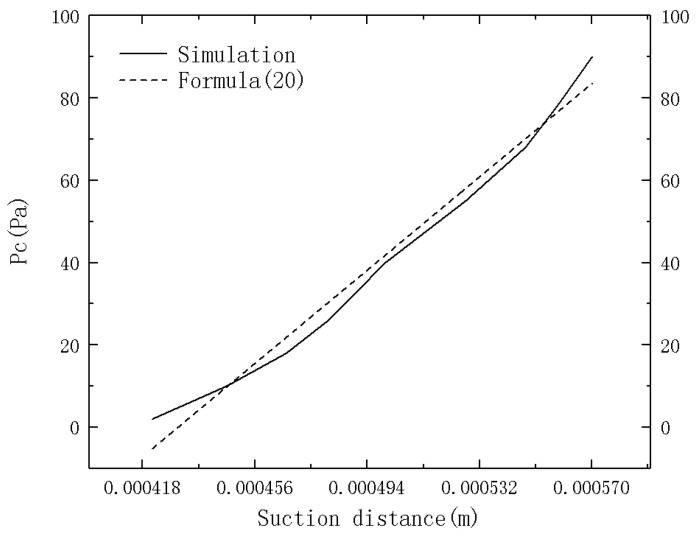
P_c_ value curves corresponding to differentsuction distance.

**Figure 8 micromachines-12-00018-f008:**
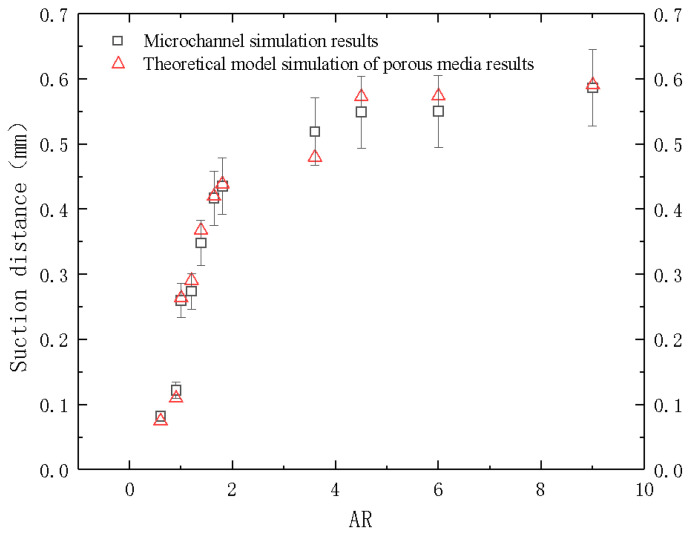
Curves of suction distance depending on AR from microchannel and porous media models.

**Figure 9 micromachines-12-00018-f009:**
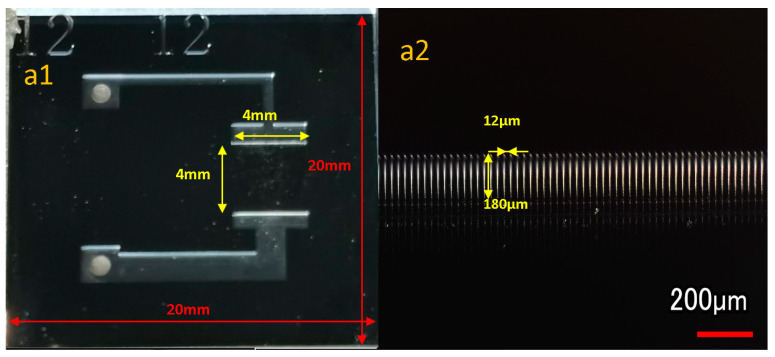
Microchannel sample images: (**a1**) Overall structure of microchannel heat exchanger; (**a2**) SEM image of the parallel microchannels array, in which each channel is 12 μm wide and 180 μm deep; (**b1**–**b4**) The images at the same magnification of the microchannel array, the depth of the parallel microchannel array is 180 μm, the width are 12, 16, 20, 30 μm.

**Figure 10 micromachines-12-00018-f010:**
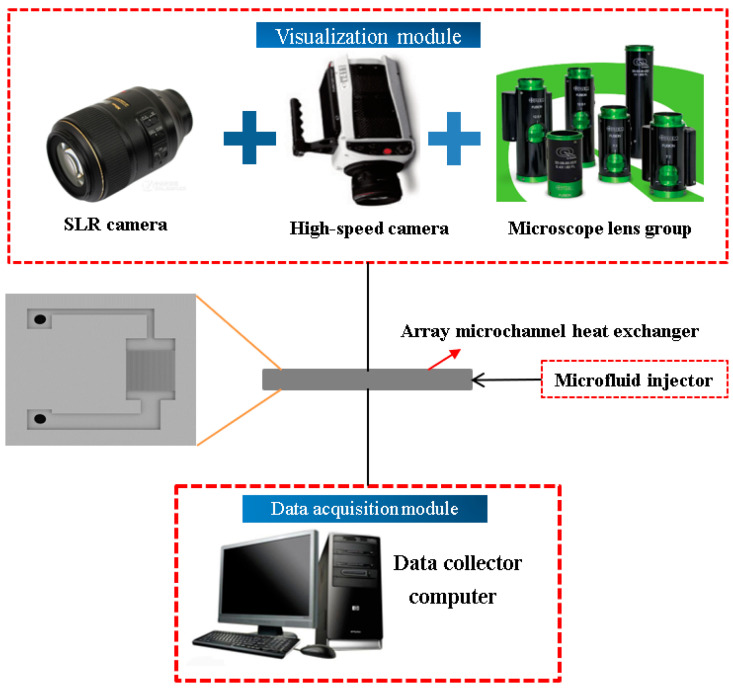
Schematic diagram of capillary suction experiment flow.

**Figure 11 micromachines-12-00018-f011:**
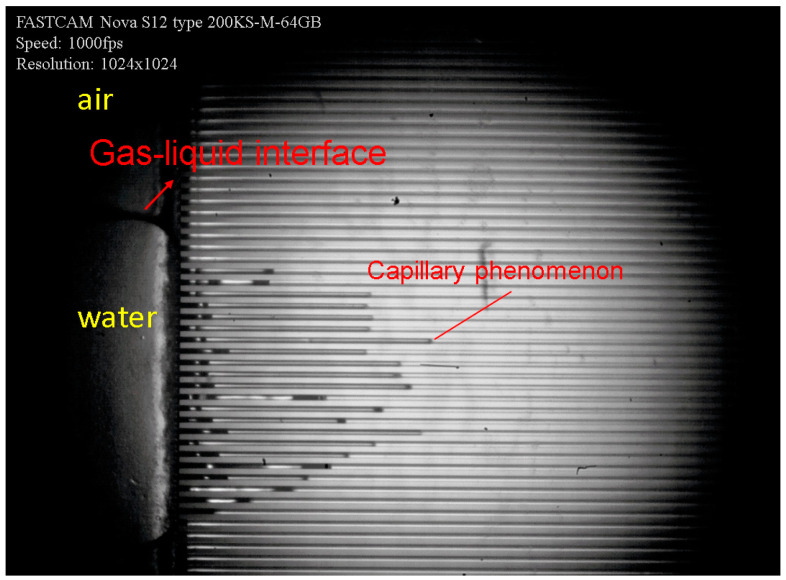
Capillary suction image: The left part of the image is the cavity at the entrance of the microchannel array, from which the liquid flows into the microchannel. The gas-liquid interface can be clearly seen in the figure. The middle part of the image is capillary phenomenon.

**Figure 12 micromachines-12-00018-f012:**
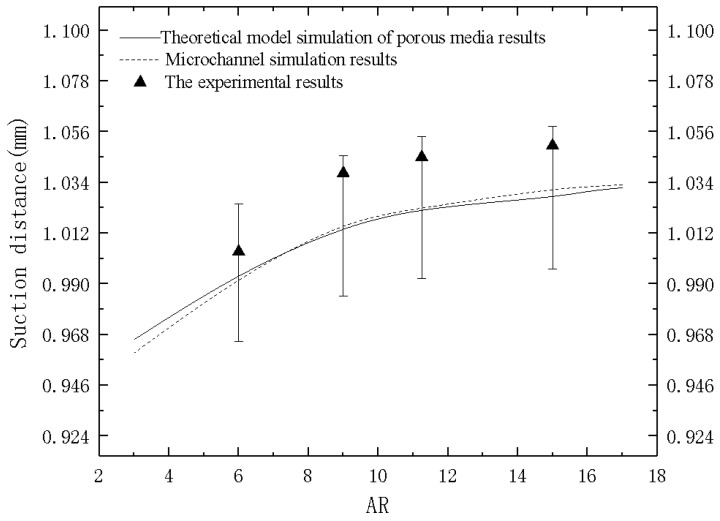
Comparison of experimental and simulation results.

**Table 1 micromachines-12-00018-t001:** Parameters of water and silicon at room temperature at 1 atm.

Variable	Density (Kg/m^3^)	Viscosity (Pa·s)	Thermal Conductivity (W/m·K)
water	998.2	1.003 × 10^−5^	0.6
silicon	2328	—	150

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
