# Peer review of "Study on Flow Characteristics of Working Medium in Microchannel Simulated by Porous Media Model"

_micromachines, 2020, doi:10.3390/mi12010018_

Round 1
Reviewer 1 Report
Thank you for submitting your manuscript in our journal. your research is focused on flow characteristics in microchannel with AF effect for understanding heat dissipation in single phase and multi-phases. you have theoretical, computational and experimental approaches to understanding phenomena with many computational, theoretical and experimental analysis. In manuscript, you mentioned about approvement of heat dissipation efficiency, but the results do not deal withs heat dissipation efficiency directly. So, your additional explanation is required to connect your result with heat dissipation efficiency. originality of your approach is not mentioned enough at introduction. Furthermore, experimental results are weak than theoretical and computational results, so additional experiments are essential. In expression, your manuscript is confused to understand because of repetitive mentions about approaches. English expression and manuscript structure comprehensively should be revised. Therefore, i give comments to improve your manuscript quality. Therefore, i suggest publication of your manuscript after major revision.
- In the introduction, the sentences should be properly distributed in paragraph for understanding. For current condition, it is difficult to understand research.
- The novelty of this study is not enough. Please explain additionally about the originality of this study.
- For the readability of your manuscript, the sentences should be written again to convey meaning clearly. I recommend to commission manuscript proofreading to professional agency.
- You need to explain detailed examples about limitation at line 97 in introduction.
- Please attach table of nomenclature.
- Mention what part of porous media is physically, and add cross sectional are of microchannel in schematics. Kindly explained schematic images is essential for extensive understanding of your manuscript.
- 1, 2.2, and 2.3 have repeatedly explained. It would be better to put the methodologies together and organize them.
- There are too many example and explanation for previous studies, so only attach essential examples to control the length of a paragraph.
- Previous studies explained about heat dissipation efficiency directly, but your research only mention about permeability, pressure, suction distance. Therefore, can you explain relation about factors you mentioned and heat dissipation efficiency? And try to draw heat dissipation efficiency if you can.
- In figure 12, there is too small number of samples to draw graph. Please extend these range from 0 to 16 at least 5. And make table to compare experimental results with theoretical model and simulation results.
- Your research is focused on AR effect for heat dissipation. Micro channel cross section SEM image is helpful to understand your manuscript.
- In experimental parts, the additional experiments are necessary for single phase and multi-phase to analysis compare with theoretical and computational results.
Author Response
Response to Reviewer 1 Comments
Thank you for submitting your manuscript in our journal. your research is focused on flow characteristics in microchannel with AF effect for understanding heat dissipation in single phase and multi-phases. you have theoretical, computational and experimental approaches to understanding phenomena with many computational, theoretical and experimental analysis. In manuscript, you mentioned about approvement of heat dissipation efficiency, but the results do not deal withs heat dissipation efficiency directly. So, your additional explanation is required to connect your result with heat dissipation efficiency. originality of your approach is not mentioned enough at introduction. Furthermore, experimental results are weak than theoretical and computational results, so additional experiments are essential. In expression, your manuscript is confused to understand because of repetitive mentions about approaches. English expression and manuscript structure comprehensively should be revised. Therefore, i give comments to improve your manuscript quality. Therefore, i suggest publication of your manuscript after major revision.
Point 1: In the introduction, the sentences should be properly distributed in paragraph for understanding. For current condition, it is difficult to understand research.
Response 1: Considering the Reviewer's suggestion, I have rearranged the sentences in the introduction
Point 2: The novelty of this study is not enough. Please explain additionally about the originality of this study.
Response 2: Considering the Reviewer's suggestion, I have added some sentences at line 86/99 in the introduction part to entrance the novelty of this study.
Point 3: For the readability of your manuscript, the sentences should be written again to convey meaning clearly. I recommend to commission manuscript proofreading to professional agency.
Response 3: Considering the Reviewer's suggestion, I have modified the sentences and grammar of the whole article.
Point 4: You need to explain detailed examples about limitation at line 97 in introduction.
Response 4: This sentence at line 97 has been deleted among the major changes. We have explained the "limitation" at line 17/78/92. It is mainly limited by the large amount of computation and the difficulty in modelling.
Point 5: Please attach table of nomenclature.
Response 5: Considering the Reviewer’s suggestion, I have added the table of nomenclature to the paper.
Point 6: Mention what part of porous media is physically, and add cross sectional are of microchannel in schematics. Kindly explained schematic images is essential for extensive understanding of your manuscript.
Response 6: Considering the Reviewer’s suggestion, I have added physical figures of the porous media in Fig 1 and Fig 5, and added schematic explanations of the images in Figs.
Point 7: 1, 2.2, and 2.3 have repeatedly explained. It would be better to put the methodologies together and organize them.
Response 7: Considering the Reviewer's suggestion, I have modified this part by putting the methodologies together and organizing them in the paragraph at line 111, and deleting others.
Point 8: There are too many example and explanation for previous studies, so only attach essential examples to control the length of a paragraph.
Response 8: Considering the Reviewer's suggestion, I have cut out the unnecessary parts and left a brief description.
Point 9: Previous studies explained about heat dissipation efficiency directly, but your research only mention about permeability, pressure, suction distance. Therefore, can you explain relation about factors you mentioned and heat dissipation efficiency? And try to draw heat dissipation efficiency if you can.
Response 9: The factors I have mentioned directly affect flow, and flow affects liquid replenishment, reflux, etc., thus affecting heat dissipation. However, although the research object of this paper is the microchannel array heat exchanger, only the flow is studied, and the heat dissipation is only the description of the research background, and there are few studies on how to improve the heat dissipation. I am very sorry for the unclear description. In addition, considering the suggestions of reviewers, I have made some explanations and modifications at line 57 and 95.
Point 10: In figure 12, there is too small number of samples to draw graph. Please extend these range from 0 to 16 at least 5. And make table to compare experimental results with theoretical model and simulation results.
Response 10: Considering the Reviewer's suggestion, I have added a set of experiments with the AR of 11.25, as well as numerical simulation results on a larger range, extending to 3 to 17. Since only four kinds of AR were processed (6, 9, 11.25 and 15), and it takes 3 months to process the new samples, the experiments of these AR samples can only be done at present. And the final experiment is only aimed to verify the correctness of the numerical simulation results. The results of numerical simulation are sufficient to draw a graph and compare with the experimental result.
Point 11: Your research is focused on AR effect for heat dissipation. Micro channel cross section SEM image is helpful to understand your manuscript.
Response 11: Considering the Reviewer's suggestion, I have added a SEM image in Fig 9.
Point 12: In experimental parts, the additional experiments are necessary for single phase and multi-phase to analysis compare with theoretical and computational results.
Response 12: This paper focuses on the establishment of mesoscopic model to simplify the simulation of microchannel array. In the theoretical study of single-phase flow, we have established the geometric model which is completely consistent with the real object, so we think that the microscopic model is correct and reliable. The theoretical and numerical simulation of single-phase flow is based on this premise, so no single-phase experiment is done. In the end, we do the experiment of two-phase flow to verify the correctness of the flow model as a whole.
Reviewer 2 Report
please see the attached file

Author Response
Response to Reviewer 2 Comments
The authors provided an interesting method to simplify the study of flow in microchannels using the model of porous media. The design of experiment is well conducted and such methods are useful to simplify the design of heat exchanger. The match between experiment and both calculations/ simulations is also convincing. On the other hand I think that the manuscript could be significantly improved (notably the clarity, conciseness, formulations...). Below a list of comments to be considered by the authors. Note that not all typos are listed here, important revisions are required.
Point 1: Line 20: please double check if an acronym (UDF) is allowed in the abstract –this acronym is only defined in line 306
Response 1: Considering the Reviewer's suggestion, I have explained the definition of UDF in the abstract at line 22.
Point 2: Line 24: please reword, the “paper “did not design or manufacture samples
Response 2: Considering the Reviewer's suggestion, I have modified it at line 25.
Point 3: Line 27: the results “can be used…” or rather “have been used to…”?
Response 3: Considering the Reviewer's suggestion, I have changed the sentence to " The simplified model of microchannel array significantly saves the computational cost and time, and provides guidance for the related experimental researches." at line 29.
Point 4: Line 56: call to ref [8] that is incomplete (in line 511), please double check if the initials of the authors are required in the text.
Response 4: I have checked the author's name is "Remco van Erp", which is abbreviated as "Erp R V".
Point 5: Line 102: -6 in exponent or simply “less than 1 micrometer”
Response 5: Considering the Reviewer’s suggestion, I have revised it at line 102.
Point 6: Line 116: conditions instead of condition?
Response 6: Considering the Reviewer’s suggestion, I have revised it and changed the sentence to “This paper simulates a silicon-based rectangular microchannel array with different aspect-ratios, equivalent diameters and inlet flow velocities using FLUENT.” At line 108.
Point 7: Line 117/118: this sentence could be reworded
Response 7: Considering the Reviewer's suggestion, I have modified this sentence to “Then we modify the porous media momentum equation through theoretical analysis to achieve the simulation of the internal working medium flow pressure drop loss.”
Point 8: Line 125/126: same remark
Response 8: Considering the Reviewer's suggestion, I have modified this sentence to “The paper using Mesh software to generate hexahedral mesh with the same size to maintain mesh quality.”
Point 9: Line 130: did you check this assumption after test? Or simply add a typical flow rate or Re number here
Response 9: Yes, we have calculated this conclusion before, and checked it in experiments. And I also add the ref17\ref18 to verify it.
Point 10: Line 132: could be useful to better define porosity which is a key element of your research
Response 10: Considering the Reviewer's suggestion, I've added a definition of porosity: “Porosity is a key parameter, which is the percentage of the pore volume in a porous medium and the total volume of the material.” at line 123.
Point 11: Legend of Fig 1 could be expanded to better describe the drawing
Response 11: Considering the Reviewer's suggestion, I have extended the legend of Figure 1 and reedited the details of the Fig. “Schematic of the single-phase flow model: (a) The brown part of the figure is a microchannel;(b) The brown part of the figure is porous media. The inlet and outlet are indicated by the arrows.”
Point 12: Line 142: “flow process” or simply “flow”?
Response 12: Considering the Reviewer's suggestion, I have changed the I have revised it.
Point 13: Line 143/144: to be reworded / simplified
Response 13: Considering the Reviewer's suggestion, I have simplified it to “The governing equations of incompressible fluid can be derived from the conservation of mass and momentum.” at line 140.
Point 14: Line 147: velocity vector in bold
Response 14: Considering the Reviewer’s suggestion, I have revised it.
Point 15: Line 148: hydrodynamic or simply dynamic viscosity of the fluid
Response 15: Considering the Reviewer's suggestion, I have modified it to “dynamic viscosity”.
Point 16: Line 151: please rework or add missing terms; could you precise slip or no slip conditions along the walls?
Response 16: Considering the Reviewer's suggestion, I have added the slip free boundary condition at line 148.
Point 17: Line 153: please put the 5 as an exponent / idem in Table 1 + m^3 for the density unit
Response 17: Considering the Reviewer’s suggestion, I have revised it at line 149 and the table 1.
Point 18: Line 156/157: please reword
Response 18: After consideration, I think this sentence is not necessity, so I have deleted it.
Point 19: simulates?
Response 19: Considering the Reviewer’s suggestion, I have revised it.
Point 20: Line 168: because this equation seems, at a first glance, not homogeneous, it could be great to add a ref and to better do the terms (e.g. C2) and the link with equation 2
Response 20: Considering the Reviewer's suggestion, I have added a ref to explain the origin of formula (3) and explained how it relates to equation (2) “Porous media are modelled by adding a momentum source term to the standard fluid flow equations”at line 158.
Point 21: Line 170: the 2 in index please
Response 21: Considering the Reviewer’s suggestion, I have revised it at line 161.
Point 22: Line 195: could be great to add units when relevant (e.g. permeability has units while porosity has none)
Response 22: Porosity is just a ratio, so it has no unit.
Point 23: Line 206: units for K? (m^2?) in Fig 2, does the interconnection line make sense ? (simulation or simply spline line)
Response 23: Considering the Reviewer's suggestion, I have added units in the Fig 2, and the line is just simply spline line.
Point 24: Line 210: “some errors” for a factor 8, it is an euphemism.. I agree that the trend seems +/- similar, but it is not so obvious because the Formula curve is not at the right scale
Response 24: Considering the Reviewer's suggestion, I have added some explanation and revised the sentences at line 192: “However, there are some errors due to the difference in channel arrangement and cross section. The aspect-ration is introduced as a parameter to correct the errors through regression analysis. The capillary permeability formula for calculating rectangular cross section microchannel beam permeability is:”. After the modification of equation (6), the scales are unified.
Point 25: Line 211: available or enables? Not clear, please mention AR in “the introduction of the aspect ratio AR”
Response 25: Considering the Reviewer's suggestion, and I have corrected it and mentioned AR in “the introduction of the aspect ratio AR”.
Point 26: Line 213: cases?
Response 26: Considering the Reviewer's suggestion, I have revised it.
Point 27: Line 215: please reword
Response 27: Considering the Reviewer's suggestion, I have revised it to “Therefore, the model can be used to simulate the flow resistance inside the microchannel.” at line 198.
Point 28: Fig 3: add units for K
Response 28: Considering the Reviewer's suggestion, I have revised it.
Point 29: Fig 4: why the point at AR=15 shows such a large error bar?
Response 29: Because the error bar is set to 15% of the base, and AR=15 has a large base.
Point 30: Line 242/245: too long sentence, difficult to follow you…
Response 30: Considering the Reviewer's suggestion, I have divided the sentence into three sentences: “In this section, the gas-liquid two-phase flow of rectangular parallel microchannel is numerically simulated. The geometric models of closed three-dimensional parallel array microchannels with different aspect ratios and porous media with porosity of 0.5 were established. The capillary force acting part is a 4mm long rectangular calculation area, which is respectively micro channel and porous medium” at line 219.
Point 31: Line 258 to 264: vectors in bold in previous eqs and now with arrows… moreover some arrows are missing (S in eq 10)
Response 31: I am sorry for these mistakes, and I've changed all the vectors to bold and checked for missing.
Point 32: Line 266: the arrows are no longer present…
Response 32: I'm very sorry, and I have corrected it.
Point 33: Line 269: CSF means continuum surface force in some papers (Brackbill notably), please clarify this terminology or use CFT for continuous surface tension
Response 33: Considering the Reviewer's suggestion, I have changed them to continuum surface force.
Point 34: Line 271: add a ref + reformulate this sentence
Response 34: Considering the Reviewer's suggestion, I have added a ref and reformulate the sentence: “When using this model, the governing equation adds a source term of the surface tension.”
Point 35: Fig 6: micrometer or mm would be a better unit choice / same remark for Figs 8 and 12
Response 35: Considering the Reviewer's suggestion, I have changed the units to mm in the three Figs.
Point 36: Line 305 is generated or takes place
Response 36: Considering the Reviewer's suggestion, the latter is right. I have corrected it.
Point 37: Eq 13 / 14: please use always the same notations for vectors!
Response 37: Considering the Reviewer’s suggestion, I have revised it.
Point 38: Line 319: could you double check the name/notation of this formula
Response 38: Considering the Reviewer's suggestion, I have used the Gaussian divergence theorem instead of the name at line 296.
Point 39: Line 335: you suggested that an existing method was adapted for your purpose, this should be clarified (+ ref shall be added in the previous section)
Response 39: Considering the Reviewer’s suggestion, I have revised it and explained the link between equation (17) and equation (18) at line 306/308.
Point 40: Line 345: fits?
Response 40: Considering the Reviewer’s suggestion, I have revised it.
Point 41: Fig 7: I guess Cp has a unit, please mention it in the Fig7
Response 41: Considering the Reviewer’s suggestion, I have revised it.
Point 42: Line 420: not clear, are you mentioning iso or anisotropic etching to get high aspect ratios?
Response 42: Considering the Reviewer's suggestion, I have made sure it's anisotropic and corrected it.
Point 43: Line 423: anodic bonding?
Response 43: Yes, I have corrected it. I am sorry for my wrong translation.
Point 44: Line 445: what do you mean by “chanciness”?
Response 44: If only one experiment is done, the accuracy of the experimental results cannot be guaranteed. Therefore, I use “chanciness”.
Point 45: Fig 12: the results are not so bad, a better scale or the use of % of error could better show this
Response 45: Considering the Reviewer's suggestion, I have used % of error in the sentence “the perspective of simulation error remained within 3%”.
Point 46: Line 463 + conclusion: in principle you are right, but I guess that fitting factors used to match simulations vs calculations are valid up to a certain extent that could be better specified here (e.g. AR range…)
Response 46: Considering the Reviewer's suggestion, I have added the AR range in the conclusion and line 469.
Round 2
Reviewer 1 Report
Although the improved manuscript is not perfect, authors comprehensively revised manuscript with additional experiment i suggested. But you need to check current manuscript again abut miss spell, unnatural expression and position of figure, graph for improvement your manuscript again before publication. I agree with acceptance of this manuscript for this journal.
Reviewer 2 Report
The authors addressed well the comments and suggestions.